# Recent Advancements and Challenges of AIoT Application in Smart Agriculture: A Review

**DOI:** 10.3390/s23073752

**Published:** 2023-04-05

**Authors:** Hasyiya Karimah Adli, Muhammad Akmal Remli, Khairul Nizar Syazwan Wan Salihin Wong, Nor Alina Ismail, Alfonso González-Briones, Juan Manuel Corchado, Mohd Saberi Mohamad

**Affiliations:** 1Faculty of Data Science & Computing, University Malaysia Kelantan, City Campus, Kota Bharu 16100, Kelantan, Malaysia; hasyiya@umk.edu.my (H.K.A.);; 2Institute for Artificial Intelligence and Big Data, Universiti Malaysia Kelantan, City Campus, Kota Bharu 16100, Kelantan, Malaysia; 3Grupo de Investigación BISITE, Departamento de Informática y Automática, Facultad de Ciencias, University of Salamanca, Instituto de Investigación Biomédica de Salamanca, Calle Espejo 2, 37007 Salamanca, Spain; 4Health Data Science Lab, Department of Genetics and Genomics, College of Medical and Health Sciences, United Arab Emirates University, Al Ain 17666, United Arab Emirates

**Keywords:** artificial intelligence of things, smart agriculture, internet of things, artificial intelligence, post-pandemic, crop diseases

## Abstract

As the most popular technologies of the 21st century, artificial intelligence (AI) and the internet of things (IoT) are the most effective paradigms that have played a vital role in transforming the agricultural industry during the pandemic. The convergence of AI and IoT has sparked a recent wave of interest in artificial intelligence of things (AIoT). An IoT system provides data flow to AI techniques for data integration and interpretation as well as for the performance of automatic image analysis and data prediction. The adoption of AIoT technology significantly transforms the traditional agriculture scenario by addressing numerous challenges, including pest management and post-harvest management issues. Although AIoT is an essential driving force for smart agriculture, there are still some barriers that must be overcome. In this paper, a systematic literature review of AIoT is presented to highlight the current progress, its applications, and its advantages. The AIoT concept, from smart devices in IoT systems to the adoption of AI techniques, is discussed. The increasing trend in article publication regarding to AIoT topics is presented based on a database search process. Lastly, the challenges to the adoption of AIoT technology in modern agriculture are also discussed.

## 1. Introduction

COVID-19 left a visible and evident economic impact on the agriculture sector. Issues such as pest attacks and bacterial infections resulted in large-scale crop diseases [1,2]. The traditional approaches must be more effective in resolving the issues, forcing researchers to consider using IR4.0 technology to tackle the problems and transform the agricultural industry.

In the post-pandemic era, smart agriculture presents itself as an appropriate solution for labor shortages and a continuous food supply chain [3,4]. Only some countries, such as the U.S. and South Korea, have established holistic visions and frameworks for smart agriculture solutions to achieve their sustainable development goals [5]. Artificial intelligence (AI) and internet of things (IoT) technologies were the most influential paradigms that played a vital role during the COVID-19 pandemic [6]. As the most popular set of technology for the 21st century, this integrated system can perform data informatization [7], efficient remote-control monitoring in real-time [8,9], and intelligent management [10].

A typical IoT system consists of wireless sensor networks (WSN) installed in different venues to collect spatial and temporal data. Such a system includes a global visualization consisting of a collection of “things” or objects or devices connected with micro-controllers, embedded intelligence, communication means, sensing properties, and actuation. All these objects are connected via internet protocol (IP) [11,12]. In industrial IoT applications, AI functions as the technology enabler for hundreds of devices and a wide range of applications, including environmental monitoring, agriculture [13,14], smart grids [15], cities [16], buildings [17], homes [18], transportation [19], and healthcare [20].

AI supports the application, implementation, infrastructure, and management of IoT technology. The application of AI in IoT environments boosts operational efficiency, provides better risk management, triggers enhanced products/services, and increases the scalability of IoT [21,22]. Fujitsu has developed a technology for estimating human body postures using millimeter wave sensors and cloud data [23]. AI techniques include image optimization, sensor processing, data transmission, and decision-making. The processes involved require high-speed data streams, low-latency communications, fast processing time, and time-sensitive actions on the IoT system [24]. Based on reinforcement learning models, the model works in the compact memory space of IoT devices [25].

Paramount quantities of IoT data require powerful AI techniques for pre-processing and preparing data to reduce noise, minimize dimensionality, and remove possible redundancies [26]. In most reports, AI techniques such as artificial neural networks, fuzzy logic, and evolutionary computation are mainly used for heterogeneous purposes [27], including classification, regression, signal processing, forecasting, decision support, and data transmission. Benefitting from their learning capabilities, various deep learning methods are frequently used in the development of intrusion detection systems [28].

### Contributions of This Study

There are still limited numbers of review articles regarding AIoT in smart agriculture, although it is widely reported for other applications. Here, we specifically focus on the AIoT concept by exploring AI and IoT as an overview. The advantages and potential challenges through a systematic literature review are also included. The contributions of this study can be summarized as follows.

(1)We discuss the AIoT concept, from smart devices in IoT systems to the adoption of AI techniques.(2)We present a systematic literature review to highlight the increasing trend in the article publication regarding AIoT in different applications… and the progress of AI and IoT.(3)We summarize a few promising applications of AIoT and other AI/IoT-enabling technologies.(4)We highlight the challenges of AIoT adoption.

## 2. Artificial Intelligence of Things (AIoT)

### Concept

The rapid evolution of AI, IoT sensors, and 5G infrastructures into robust technologies has led to their intersection under the paradigm of AIoT (artificial intelligence of things) [29]. Despite AIoT still being in its infant stage, the applications and trends that it encompasses are reshaping the future of enormous business potential. Overall, the IoT system provides data flows and is further utilized with AI techniques to integrate the data, interpret the data, perform automatic image analysis and data prediction, etc. Figure 1 illustrates the integration of AI with IoT for better user outcomes [30].

In agricultural applications, the integration of AIoT mostly regards controlling harvests, greenhouse parameters, and smart fertigation to induce reactions to any change in external conditions. For instance, convolutional neural networks are adopted to predict and detect possible diseases in applications at any scale on the basis of the collected IoT data [31].

AIoT makes the operation and management of agriculture easier to access, effective, and autonomous for the users. AIoT will become one of the main driving forces for smart agriculture. However, some barriers still must be overcome, such as the cost factors and the readiness for technology adoption as standard practice.

Several promising AIoT applications are discussed thoroughly in this paper. The motivation is triggered by existing proven applications of AI/IoT, such as computer vision, video surveillance systems, and infrared cameras. The review propounds ideas on further improving AIoT applications and advantages that benefit the users. The challenges in technology adoption are also discussed.

## 3. Methodology

A systematic literature review (SLR) approach was applied to select, review, and critically appraise existing published articles with similar research problems before summarizing selected primary articles. The overall stages of the SLR consist of planning an reviewing, a similar approach to that reported previously. Figure 2 presents the step-by-step of the SLR conducted in this study; a similar approach was applied previously [32].

At the planning stage, the research objective and questions were identified and finalized before developing and evaluating the review protocol. Then, the identification of main articles using search strategies, the selection of articles, and the extraction of the data were progressed during the review stage. Finally, the summary and report were made after the data were synthesized and interpreted.

### 3.1. Objective

The objective of this study is to review the current progress of AIoT adoption in smart agriculture and to identify the research gap and potentials related to AIoT technology. To the best of our knowledge, few articles related to AIoT applications in smart agriculture have been published, hence the scope of the study focusing on the technology based on deep learning, computer vision, and intelligent video surveillance systems. Table 1 presents the research questions of this study.

### 3.2. Process

The automatic database search was carried out on 12 December 2022 with the search keywords “Artificial Internet of Things (AIoT)” and “AIoT in agriculture”, as shown in Table 2.

### 3.3. Article Selection Criteria

This section focuses on the selection process of the articles and specific criteria applied to the available filters of each of the databases. Table 3 presents the summary of inclusion and exclusion criteria. The contents of each article were read, reviewed, and considered during the process. A total of 11 articles remained, all of which were in the context of AIoT applied in smart agriculture.

### 3.4. Quality Assessment and Data Extract

After articles were identified, quality assessment was carried out to ensure the article’s quality and its relevancy to this study. Table 4 shows five questions to comply in order to assess the quality of the articles based on specific scores: yes (1.0), partially (0.5) and no (0).

The total score for each article from the sum of the values obtained from answers. A score of 1.0 indicated well-matched with this study, and 0.0 indicated otherwise. A cut-off score of 0.5 was defined as moderate, and only articles with a score greater than 0.5 were considered for this study.

All primary articles that passed inclusion and exclusion criteria in previous stage (Section 3.3) were read and evaluated to acquire the score. Table 5 shows the summary of scores from the quality assessment of selected 11 articles as the input set, of which 3 articles received scores less than or equal to 0.5.

## 4. Results

At the primary stage, the initial amount of five hundred and sixty articles was found from three databases. Eleven articles were considered as final primary articles after considering inclusion and exclusion criteria. It cannot be neglected that the biases during the initial stage of primary article selection due to the small number of articles were published that according to similar scope of study.

This may also imply that the possibility of irrelevance of the articles to answer the research questions. Additionally, the choice of databases also cannot be considered can represent the completeness of the studies. Other reasons that can also be considered are: (a) the limited AIoT applications in agriculture have been reported; (b) related policies and mechanisms also might be under-developed to push as the assessment. However, it is worth noting that the number of articles reporting on AIoT adaption in smart agriculture has increased massively since 2021, from less than 5 in 2017 to a projection of close to 50 papers in 2023 (Figure 3).

### 4.1. Current Applications of AIoT in Smart Agriculture

The articles presented in the next subtopics have been contributed to the massive research development of AI and IoT in smart agriculture and have potential for AIoT application in the future. To date, 37,230 articles have been published regarding the scope of AI and IoT in agriculture (Table 6).

#### 4.1.1. Deep Learning Methods

Deep learning offers many opportunities in various agricultural stages, and its application has increased substantially in recent years, with numerous studies providing innovative insights into this topic. Table 7 shows the recent work on deep learning, and Table 8 shows that for traditional machine learning, both for the agriculture application. Most proposed works focus on predicting and classifying crop production, disease detection, and automating irrigation systems. The data consists of various sources, including irrigation systems, temperature, humidity, water, weeds and crops, disease, and fruit grading. Combining IoT data with advanced deep learning algorithms has brought vast development and continuous breakthroughs in agriculture, AI, and deep learning fields.

A recent study used a hybrid method based on empirical mode decomposition (EMD) to decompose the data. At the same time, a gated recurrent unit (GRU) was utilized as a predictor to predict various climate data, including temperature, humidity, and wind speed. The performance of the proposed method is superior and can produce the highest accuracy in precision agriculture production. Another predictor model for automating normalization using reservable automatic selection normalization (RASN) based on indoor data from the smart greenhouses is presented in [39]. The authors have introduced scaling and translating techniques with parameters that can be learned to improve prediction and adaptability in innovative agriculture systems. Combining IoT devices with the DL algorithm is a challenging problem due to device limitations, as DL requires high computational resources to process the IoT data. To solve this issue, ref. [40] proposed the deep learning based on fog nodes (DLEFN) algorithm and experimented with it in order to determine the optimal layers in DL for execution in each fog node. The availability of device capacity and bandwidth influences the results. The output shows that the proposed algorithm can reduce network congestion and utilize resources efficiently.

Several CNN algorithms have been improved and utilized in computer vision for smart agriculture. For example, ref. [41] introduced a computer vision monitoring system to monitor tomato growth stages using a regional-based convolution neural network (R-CNN). Other utilized algorithms include artificial neural networks (ANN), KNN, and support vector machine (SVM). The proposed system shows the highest accuracy in predicting flower and fruit and in maturity grading. The RCNN was also used to detect leaf disease in smart agriculture [42]. The authors combined R-CNN with MASKRCNN to detect the infected disease in apple leaves. Transfer learning was used to extract features from the pre-trained CNN model. Results confirmed that the proposed framework was superior compared to other frameworks. Aside from that, long short-term memory (LSTM) is a recent DL technique that has had remarkable results.

On the other hand, auto-encoder (AE) is an artificial neural network that shows excellent performance in learning codings for unlabeled data. Both LSTM- and AE-based techniques have been used in smart agriculture. In a study [43], a deep neural network based on LSTM was used in an IoT irrigation system for precision agriculture (DLiSA). In their work, LSTM was utilized to predict several irrigation conditions including soil moisture, irrigation time, and water amount needed to spray the arable land. The simulation of the proposed system showed that it was efficient in water usage compared to other state-of-the-art irrigation systems. The authors of [44] focused on an unmanned aerial vehicle (UAV) framework in which several techniques—including blockchain for data authentication, sparse auto-encoder (SAE) for data transformation, and stacked long short-term memory (SLSTM)—were combined for training, and they evaluated the result. The experiments demonstrated that the framework outperformed other state-of-the-art block chain and non-block chain frameworks. Deep reinforcement learning is another advanced deep learning model used in smart agriculture [51]. Specifically, the model was combined with cloud computing technology, resulting in an intelligent model that can determine the amount of water needed for irrigation. The work resulted in increased production and an improved crop growth environment.

#### 4.1.2. Computer Vision

A smart agriculture system using DL-based computer vision is promising due to the massive growth of agriculture data commonly collected from IoT sensors. Other researchers also studied and applied traditional machine learning (also in combination with DL) techniques, such as fuzzy logic, SVM, supervised learning, decision tree, linear regression, and KNN [52,53,54,55,56,57,58,59,60]. The authors of [52] proposed a small-scale agriculture machine to irrigate and weed automatically in the cultivated area. In their work, the fuzzy logic controller was applied to provide data on the wet distribution area of surface soil (WDAS). The results showed that the machine achieved an accuracy of at least 90% in weeding and watering the deep soil. In recent works from [53,54], the combination of ANN, Gaussian curve fitting, and CNN was proposed in crop farming specifically for fruit-grading systems. The SVM classifier is mainly used for various classifications in smart farming. In [55], the authors proposed an IoT framework using machine learning algorithms, as illustrated in Figure 4. The primary purpose of the framework was to measure and predict crops using supervised learning. The dataset consists of 200 instances and eight attributes. The experiments showed that the proposed model could predict the high-yield crop in precision agriculture with an accuracy of 98%. Other frameworks have been developed to detect and automate agriculture, including water supply and crop/weed spraying.

The framework commonly collects primary data from scalable IoT sensors [56,57,58]. Some works focused on the security aspect of IoT systems, specifically in performing smart agriculture—for example, ref. [56] studied risk mitigation in smart irrigation based on an intrusion detection system (IDS). The authors proposed a framework using SVM as a classifier, linear regression, and random forest based on the NSL KDD data set. The framework performance was compared to state-of-the-art machine learning algorithms based on recall and precision. Another work also used a random forest classifier in detecting crops/weeds for real-time variable-rate spraying [57]. Their work aims to improve spray yield and protect crops from disease according to a specific amount of agrochemicals and field/crop requirements. The random forest classifier detected and classified various weeds and crops. The desired amounts of agrochemicals were then sprayed using a vision-based feedback system. The simulation result shows that the proposed method is effective.

Handling enormous amounts of data is a daunting task and results in higher costs, especially if the data come from heterogeneous information. To cater to this challenge, [58] introduced a method of managing heterogeneous agriculture datasets (biological, sensory, and physical values). Various machine learning techniques have been utilized to suggest the best effort and investments in data management. Another work proposed an intelligent method based on wrapper feature selection combined with the PART classification technique (WPART) to improve crop productivity and drought prediction [59]. In their experiments, five data sets were used to train the model. Using productivity crop data including soybean and sugarcane, the proposed method could obtain high accuracy and outperform other machine learning algorithms in crop productivity and drought prediction. Machine learning and deep learning algorithms consist of various techniques and methods. Applying all these algorithms to perform prediction and classification in smart agriculture is time-consuming, challenging, and requires enormous cost. To facilitate the selection of the best machine learning and deep learning algorithms, ref. [60] studied various algorithms to be applied in smart farming. Their work guides other researchers and farmers in cultivating crops efficiently and helps improve productivity while keeping costs low. Furthermore, the work also helps farmers better manage their crops and suggests a smarter harvesting process.

It should be noted that the methods presented rely on machine learning and deep learning algorithms, which have seen substantial improvement and breakthroughs in smart agriculture. Although deep learning has received great attention in recent years, machine learning methods based on basic algorithms, such as random forest, support vector machine, and K-nearest neighbors, are still useful because of the labelled data and supervised learning [61,62]. However, due to the heterogeneous and variety data sources from the agriculture field, verification and validation of the machine learning system must be further analyzed. This area is also known as the formal method, which is a field of study that examines and strictly verifies machine learning systems both in hardware and software systems. Several approaches have been proposed to provide verification framework for the high dimensionality, complexity, and uncertainty of machine learning algorithms [63,64].

#### 4.1.3. Intelligent Video Surveillance Systems

Intelligent video surveillance systems are widely used in many applications, such as crime prevention, security, monitoring and controlling essential infrastructures, and the smart agriculture industry [63,64]. From 2010 until 2019, there were 220 video surveillance system (VSS) studies, which highlights the continuous relevance of research in this field [64]. VSS has garnered great interest in the past decades, especially with the integration of computer vision, image processing, and artificial intelligence capabilities. Figure 5 shows the VSS architecture that includes components such as sensors, servers, and network types [65]. The analytics cover the part of processing algorithms to ensure good surveillance.

With traditional video, human energy and input are required to watch and analyze the video multiple times, which is tedious and time-consuming. Many academics recently proposed various intelligent surveillance methods that can recognize human action based on high accuracy and efficiency. Intelligent VSS may consist of a few stages, such as video preprocessing, object detection, activity detection, recognition, and action classification.

The combination of AI and IoT sensors makes video surveillance smarter and recognizes an object automatically [66]. The intelligent VSS system operates without human control or within autonomous systems; it detects, analyzes, and anticipates events or behaviors of things that trigger a warning. The system can perform face recognition, identify intrusion and anomalies and send alerts to farmers to take appropriate action. The integration with the IoT system effectively manages all devices such as cameras, smoke detectors, and audio sensors in the monitoring area [66]. In the future, AIoT will allow all devices, especially VSS, to think and decide on their own if any unexpected events occur on the farm. With the fast evolution of IR4.0 Technology, video surveillance systems are also undergoing continuous development in terms of the devices and analysis based on AI algorithms. Among various AI models used for surveillance analysis are CNN, auto-encoders, and their combination. The author summarized AI methods most commonly applied in visual surveillance systems: deep learning, Gaussian, support vector machine (SVM), fuzzy logic, and nearest neighbor [65].

Meanwhile, the capabilities of smartphone technology also revolutionize farming activity. Current smartphones are equipped with various in-built sensors, such as GPS, temperature, accelerometers, and light sensors. The authors of [67] reported on the state-of-the-art of current publication on the application of smartphone sensors in agriculture. With users of smartphones being everywhere and widely connected through networks, there is an increasing opportunity in the development of smartphone-based sensor systems for agriculture. As shown in Figure 6, the sensors of smartphones monitor the environment and collect and process the data either in the smartphone itself or on local machines or remote cloud servers [68].

The main advantage of a smartphone-based sensor system (SBSS) over a typical wireless sensor network is its low cost of equipment; the built-in sensors on smartphones reduce the cost of software and central equipment [69]. For light use applications, smartphones can partially process the data with free and powerful development tools. For instance, Android smartphone users may program in Java on Android Studio Integrated Development Environment (IDE) [70]. Smartphones have become powerful tools in the agricultural sector [71], and recently, a mobile vision system was reported for identifying early-stage diseases in plants [72]. By taking photographs of the parts of the plant to analyze for the infection, these photographs are then pre-processed and transferred to distant labs. Additionally, a color estimator on the application of smartphone is reported for chlorophyll estimation [73].

However, there are still some limitations on the use of smartphones. For example, the sensor-collected data might not be complete or uniform. In addition, the quality of smartphones also varies with different levels of data accuracy are provided. The privacy of smartphones users also contributes to the need for proper protection mechanisms in order to avoid the leaking of the user’s personal data.

The use of smartphones helps farmers to be more productive and helps better decision-making after obtaining useful insight from agriculture data. The use of AI and IoT in agriculture technology has proven to be a significant improvement in farming efficiency, cost, and crop yields. However, further research and development in AI and IoT towards AIoT should be accelerated to fully optimize the future of agriculture technologies. In the future, the increasing number of smartphones and their sensor capacities will be improved and should contribute to faster adoption of AIoT in real life. To achieve this goal, it is necessary to deepen the research on whole system components improvement, sensors, data exchange and gathering, and data security [74].

### 4.2. Advantages of AIoT in Agriculture

According to past studies, about 75–80% of people live in rural areas and still depend on agriculture. In the context of AIoT, the technology can serve as a solution to help farmers increase yield and productivity exponentially. It uses various sensors connected to the internet and integrated into the satellites.

Smart agriculture, or smart farming, is the concept of maintaining and monitoring farms using modern technologies to increase the quantity and quality of products. With the advancement of sensor technologies, miniaturization, and cost reduction, farmers have access to navigation, soil scanning, data management, and pest detection technologies. AIoT-enabled devices, real-time data collection, and automation can vastly improve the smart agriculture industry. Smart farming is not the same as embedded systems, which allow AIoT to enable devices and take advantage of AI technology. Considering the benefits of IoT and AI, better possibilities could be expected from AIoT in performing better than previous technology. The subtopics that follow describe the four critical advantages that AIoT is expected to contribute to the agricultural sector.

#### 4.2.1. Disease Identification and Plant Monitoring

Farmers and researchers may use crop imagery analysis to spot early infection, enabling them to take action before the illness becomes a significant issue. One of the main advantages of employing deep learning is the ability to detect infections in their early stages. It can take a lot of time and effort to discover disease using traditional approaches, such as manual examinations by trained workers. Additionally, these techniques might only sometimes be reliable because it can be challenging for human inspectors to spot minute evidence of infection. On the other hand, deep learning systems can swiftly and reliably assess photos of crops, enabling the early diagnosis of illnesses. The system may be trained to accurately recognize infection symptoms using a sizable photo collection. This can assist farmers in acting quickly to stop the virus from spreading in order to safeguard their crops.

Crop diseases represent a significant threat to agricultural output. Frameworks for deep learning utilize sick leaves’ unique characteristics to detect different diseases. The paper employed complete deep learning (CDL) architecture to offer a multi-crop disease detection model capable of categorizing crop diseases irrespective of crop type [75].

Agricultural progress is constantly aided by early disease diagnosis, classification, and analysis of ill leaves as well as the identification of viable treatments. The authors concentrated on detecting, classifying, and predicting several plant diseases, especially in tomatoes and grapes [76]. An algorithm based on deep learning was used to extract visual features to discriminate sick leaves from healthy ones, as illustrated in Figure 7.

Identifying plant diseases is highly significant in agriculture to boost crop output. Due to recent breakthroughs in imaging, visual plant disease analysis is now used to address this issue. Authors examined the difficulties in visually recognizing plant illnesses for disease diagnosis [78].

Plant disease images are more likely to have randomly dispersed lesions, various symptoms, and complex backgrounds than other typical types of photography, making it more challenging to capture distinguishing information.

The behavior of the soil varies due to alternating climatic conditions. Pests are another major worry. Image processing has evolved into a valuable instrument for the early detection and diagnosis of plant diseases. Several methods have been employed to identify diseases in their earliest stages, resulting in little crop loss and good crop quality. The authors’ study on banana crop diseases and their potential solutions aids faster detection and diagnosis [79]. Thanks to an IoT system, agricultural data can be collected while AI mechanisms train and automatically analyze data in real time.

#### 4.2.2. Intelligent Farm Machinery and Crop Management

Complex mathematical models called “deep learning” are modelled to resemble the structure and operation of the human brain. These neural networks are trained on enormous datasets and can learn and make predictions or judgments based on the data.

Additionally, deep learning may be utilized to create autonomous agricultural equipment that can plant, weed, and harvest crops autonomously, without human assistance. This may lower labor expenses and boost overall effectiveness. Deep learning algorithms may be used to train autonomous farm machinery to navigate and operate in complex agricultural situations while making judgments based on the information obtained from sensors and other sources.

Numerous challenges remain, such as the shortage of workers in the agricultural sector and the increased demand for newer high-tech advanced machinery. New technologies within autonomous robotics are expanding in the agricultural industry. Huge investments are being made to develop autonomous agricultural mobility robots; as a result, modern farms have high prospects for increased productivity. Due to the complexity and diversity of the agricultural work environment, it is difficult to overcome current obstacles using the present machinery design. The paper examined a technique for creating and managing a mobile robot platform that may handle these difficulties in a greenhouse [80]. The developed platform has two driving wheels and four casters that could operate on a route and a rail. In addition, it provides technology for a multi-AI deep learning system to operate a robot, a robot-operating algorithm, and a VPN-based network and security communication system [81].

Food shortages are expected to worsen because of the growing world population. Due to the diversity of orchard conditions and tree types, fruit farming requires substantial manual labor, causing mechanization and automation to fall behind [82]. Mechanization is important to improve efficiency and reduce dependence on manual labor. A system for automated fruit picking using robots fitted with robotic arms was proposed by the authors of [83]. Before putting end-effectors into the fruit’s bottom half, the fruit-harvesting robot uses sensors and computer vision to identify and estimate the fruit’s location. Experiments demonstrated that this technology could detect pears and apples in the field and pick them autonomously [84].

For agricultural automation, the exact distribution of liquid fertilizer and pesticides to plants is a critical activity in precision agriculture. It provides a more cost-effective and ecologically friendly alternative to conventional, non-selective spraying by identifying and decreasing the number of chemicals used. Spraying with precision involves the detection and tracking of each plant. Traditional detection or segmentation techniques lump all plants inside an image collected by a robotic platform irrespective of the plant’s unique identifier. In addition to recognizing each plant, it is vital to track each plant to administer pesticides precisely once to each plant. The previous researcher proposed a multiple object tracking (MOT) technology that recognizes and tracks lettuce concurrently, only spraying plants that have never been treated before [85]. The approach leverages YOLO-V5 for identifying lettuce and includes plant feature extraction and data association algorithms to monitor all plants successfully.

The authors created a virtual simulation setting by fusing a robot operating system (ROS) to illustrate the possibilities of a simulator channel to present a case study on indoor robotic farming [86]. The paper developed a technique for evaluating the harvest of sweet peppers. The method uses aerial robotics control and trajectory planning, followed by deep learning-based recognition and a clustering algorithm for fruit counting. This case study illustrated that a complex robotic system may be modeled by integrating real-time modeling with almost practical rendering capacities.

#### 4.2.3. Efficient Agricultural Data Analysis

A wide variety of variables crucial for effective crop management may be predicted using deep learning. Deep learning algorithms may be used to assess sensor data and data from other sources to forecast agricultural production, soil health, and other elements that are crucial for effective crop management. This information may be utilized to optimize irrigation, fertilization, and other farming practices to enhance crop health and yield. The health and quality of crops may also be predicted using deep learning algorithms.

For instance, using information from cameras and other sensors, algorithms may be taught to find illnesses or pests in crops. This data may be utilized to spot issues early on and take appropriate actions before the problem causes serious harm to the crops. Deep learning algorithms may also forecast customer preferences and market circumstances. Farmers and other agricultural stakeholders may utilize this information to better guide their choices on what crops to grow, when to plant them, and how to market and sell them. Deep learning may assist related stakeholders in making better decisions and increasing the productivity and sustainability of their operations by enabling the construction of more precise and complex prediction models.

The work outlined the construction of a low-cost and low-power wireless sensor network (WSN) based on photovoltaic (PV) sensor nodes that can obtain ambient circumstances and soil data [87]. Soil moisture sensors are the most critical installed sensors due to huge costs and difficulties in installation, reliability, and calibration. This article presented a deep learning (DL) technique using long short-term memory (LSTM) networks to simulate a soil moisture sensor using data collected from the other transducer mounted on the node. To confirm the usefulness of the suggested soft sensing technique, this study evaluated the performance of virtual sensors and compared it to other approaches.

Recent advancements in remote sensing using unmanned aerial vehicles (UAVs) for precision agricultural operations have significantly enhanced crop health and management. UAVs outfitted with sensors, cameras, LIDAR, and thermal cameras have been used for crop remote sensing since they offer new methods and possibilities. The article examined the use of UAVs for pest and disease control, yield estimation, phenotypic measurement soil moisture assessment, and nutritional status evaluation in the sugarcane industry to boost efficiency and maintain an ecological state [88].

Proper land use and crop maps obtained from remote sensing provide critical and timely information for agricultural monitoring on a wide scale. Due to their limited model transfer capabilities, the bulk of existing multi-crop products for complex agricultural landscapes centered on standard machine learning approaches must be improved for large-scale agricultural management. That is why developing a segmentation and classification model with spatial and temporal transfer across regions and years is essential. A study developed a deep learning technique for large-scale land use and crop mapping by combining feature fusion with the up sampling of small data using the UNet++ architecture. The method improved classification accuracy for datasets with variables by analyzing the full confusion matrix [89].

In smart agriculture, computer vision and AI can optimize agricultural output while minimizing resource use and improving environmental and economic results. This effort aims to develop cutting-edge algorithms for image-based crop evaluation to help growers make real-time choices. The previous paper made two substantial algorithmic advances. First, the report devised a technique for segmenting cranberry instances which offers a number of sun-exposed berries susceptible to warming [90]. The second algorithmic contribution of an end-to-end differentiated network is a combined in-field prediction of sun irradiance and berry temperature. The integrated system evaluates the risk of overheating impacting irrigation decisions.

Developing and using new technology to solve agricultural issues and boost agricultural productivity is necessary. The authors described heterogeneous data management for agriculture to explore IoT [91]. The article proposed an IoT-based smart farming prediction and intelligent agricultural analytics model, as well as a decision tendency that reliably anticipates crop yield using a deep learning approach. Ensemble voting increases the agricultural enterprise’s profitability, efficiency, and sustainability in this model.

Another author proposed a decision system able to predict the crop yield at the country level [92]. The results calibrated and trained regression methods for the simulation model using meteorological, soil, crop, and agro-management data.

The results show that the three proposed machine learning models fit well the crop data with a high accuracy R2 and minimum values of the root mean square error (RMSE) and mean absolute percentage error (MAPE) [92].

#### 4.2.4. Weather Forecasting System for Quality Production

Deep learning can be used to anticipate weather patterns and other meteorological phenomena that may influence crops and farming activities. Deep learning algorithms may be trained to create precise and trustworthy weather forecasts by studying data from weather stations, satellites, and other sources. Farmers and other agricultural stakeholders may utilize this information to guide their decisions regarding the best times to plant, water, and harvest crops.

The chance of rainfall and the anticipated amount of precipitation may also be predicted using deep learning algorithms. This information may be utilized to improve irrigation schedules and to minimize overwatering or underwatering of crops. Additionally, deep learning algorithms may be used to forecast the possibility of catastrophic weather occurrences, such as hurricanes, floods, and droughts, which can significantly influence crops and agricultural operations.

Additionally, forecasts concerning long-term weather patterns, such as trends in temperature and precipitation, may be made using deep learning systems. Farmers and other agricultural stakeholders may make plans and decisions about what crops to grow, how to save water and other resources, and how to adapt to changing conditions using this knowledge.

Variations in precipitation adversely affect agricultural yield and inflict climatic severe conditions. Especially in rain-dominated countries, research concentrating on climate swings, such as variations in rainfall and temperature, is essential. It is challenging to predict precipitation accurately due to its dynamic nature. Using 30 years of climate data, the authors planned to develop a prediction model based on an optimized GRU neural network [93]. The minimal loss reached by the model revealed the validity of chosen elements to correctly predict precipitation irrespective of the volatility of meteorological conditions.

The publication describes field-transportable rain forecasting equipment that can identify the chance of precipitation by detecting relevant atmospheric variables, such as temperature, humidity, and atmospheric pressure, as well as sky conditions [94]. In addition to the device’s portability, it can be used to predict the likelihood of precipitation at a specific location by integrating the prediction model of cloud images using deep learning [95].

The research by authors in the preprocessing phase employed wavelet decomposition. The implementation of a long short-term memory (LSTM) network and of a suitable forecast improvement phase was further optimized by algorithms for monthly rainfall forecast modifications [96]. The approach was executed at four weather stations and compared to transfer function models, multiple regression, and other prediction strategies.

One effective strategy for conserving water is utilizing rainwater to its maximum capacity. It is possible to use weather forecasting to save irrigation water; however, unnecessary watering and yield loss should be avoided due to weather forecast unpredictability. A deep Q-learning (DQN) irrigation decision-making system based on short-term weather forecasts was established to determine the optimal irrigation method. The DQN irrigation method showed strong generalization capability and may be used to make irrigation decisions based on weather forecasts. The DQN irrigation strategy of learning from previous irrigation experiences and uncertain weather forecasts mitigated the risks associated with faulty weather forecasts [97].

### 4.3. Challenges in Technology Adoption

AIoT is very promising as it has an immense potential to transform society and business positively. According to [98], the use of the newest technology with no experience and digital skills is currently the biggest challenge in many industries since many companies have started to use “things” to connect with others [99]. As “things” connected to the internet are rapidly growing in number, it may lead to several issues in technology adoption.

One of the issues most companies face is complexity. This complexity refers to the interrelation of connected devices with other systems from a cyber-physical system perspective [100]. In technology adoption, the major challenge faced by users is the monetary budget for a given purpose, such as purchasing the tools and main training the overall system. The limited knowledge and awareness also caused challenges in technology adoption.

The other challenge is the need for more trust in AIoT technology. Lack of confidence in AI and IoT also could delay technology adoption. It is expected that AIoT technology is reliable and highly dependent on accurate knowledge in the context of the flexibility of data handling. Farmers are not tech-savvy, so they may entirely depend on the experts to understand and analyze the AIoT system. It can become challenging for them to learn and adopt the technology. Additionally, the industry found it challenging to hire experts and professionals with digital skills to implement new systems as well as operate and maintain new technology operations.

Other than that, privacy and security issues also cause a delay in technology adoption. The integration of AI and IoT can create new security risks, such as data breaches and cyber-attacks. As IoT devices collect and transmit vast amounts of data, privacy concerns are becoming more significant. Hence, organizations should be aware of the applicable laws and regulations regarding the data preservation.

In many industries, technology infrastructure was found to be an essential factor to ensuring the managerial success of the new technology [98]. Without a proper infrastructure, the company can be addressed as having an outdated technology of AIoT.

The most challenging part of deploying an AIoT model in agriculture is ensuring it maintains its performance in any environmental conditions or uncertainty factors, such as rainfall, humidity, sunlight, temperature, and water availability.

AIoT systems require many data from agriculture to produce robust AI models. In the case of agriculture, the data are usually based on seasons even though spatial data can be collected in real-time. This limitation may make it challenging to achieve the accuracy of AI models at a particular time. Moreover, an inaccurate prediction may entail costs. Errors in making predictions and recommendations could mean losing crop production for an entire year, affecting farmers’ lives and global food security. Hence, using a small portion of the farmers’ land for data analysis based on AI systems may become a part of strategic planning performed prior to deploying the AI model on the whole farm.

## 5. Conclusions

This study presented a systematic literature review of AIoT studies to highlight the increasing attention on this technology. It is worth noting that the number of articles reporting on AIoT adaption in smart agriculture has been increased massively since 2021, from less than 5 in 2017 to a projection of close to 50 papers in 2023. From IoT to the AI techniques, this brings remarkable progress towards AIoT in various applications. AIoT-enabled devices, real-time data collection, and automation are important criteria that can improve the smart agriculture industry. The current application of AI/IoT technologies with some advantages of AIoT was summarized in this paper. Several advantages of AIoT in agriculture, such as disease identification, smart farm monitoring, and efficient agricultural data analysis, were also highlighted. It can also be noted that there is some limitation on technology adoption in the smart agriculture industry as related policies and mechanisms are still under-developed to push as the assessment. Lastly, technology complexity, privacy and security issues, and under-developed infrastructure were identified as a few challenges of the application of AIoT in smart agriculture.

## Figures and Tables

**Figure 1 sensors-23-03752-f001:**
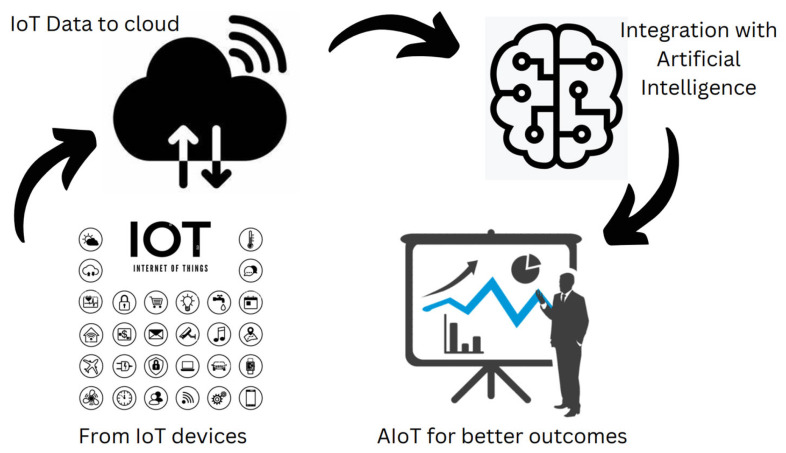
From smart “Things” in IoT systems to the adoption of AI techniques.

**Figure 2 sensors-23-03752-f002:**
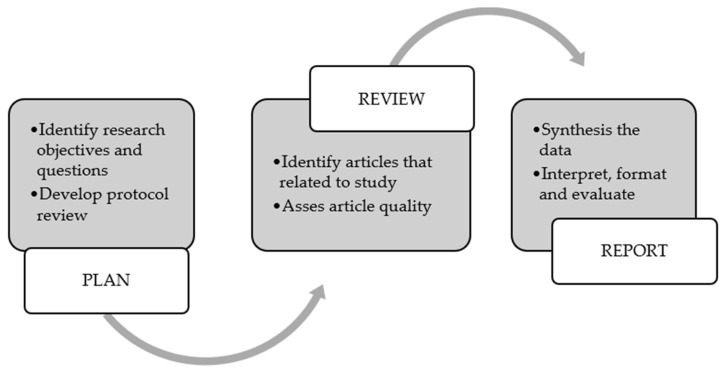
Systematic literature review approach [32].

**Figure 3 sensors-23-03752-f003:**
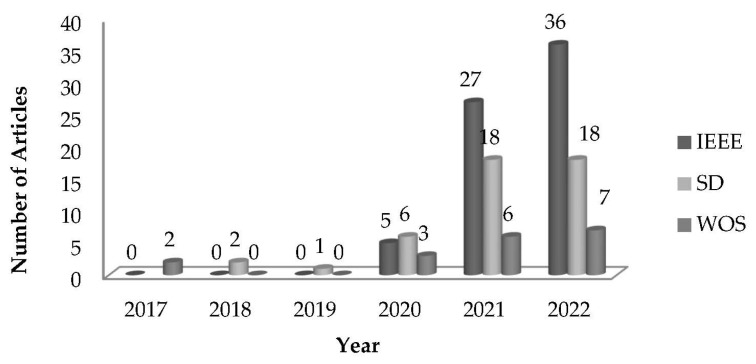
Number of articles published since 2017 related to this scope of study.

**Figure 4 sensors-23-03752-f004:**
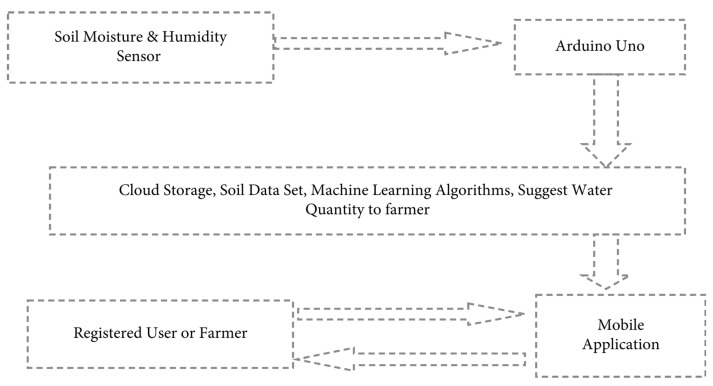
A framework for smart irrigation [55].

**Figure 5 sensors-23-03752-f005:**
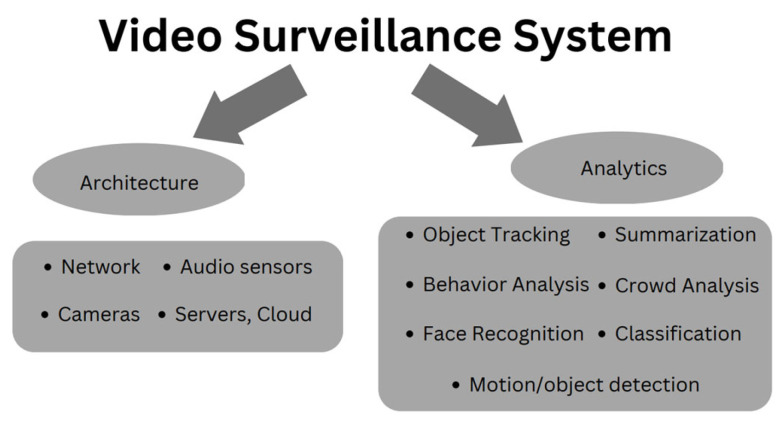
The component and functionality of VSS.

**Figure 6 sensors-23-03752-f006:**
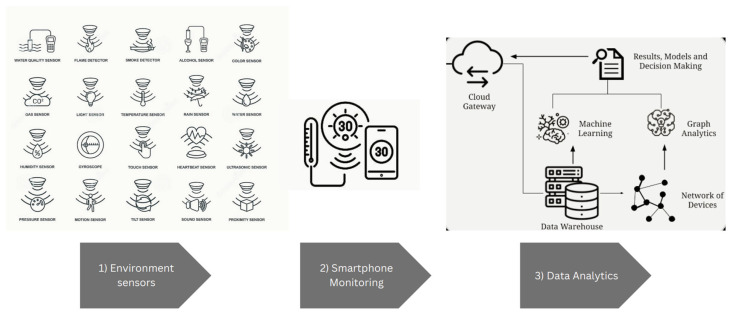
Smartphone for monitoring.

**Figure 7 sensors-23-03752-f007:**
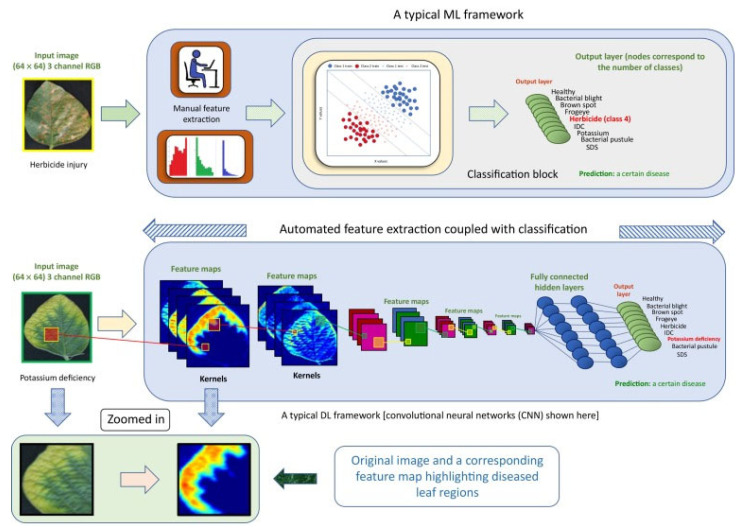
Identification of plant leaf diseases using CNN [77].

**Table 1 sensors-23-03752-t001:** Research question and justification.

ID	Research Question	Justification
RQ1	What are common subject areas of AIoT application that have been reported recently?	Providing the context and analyzing the current trends of subject area.
RQ2	Which deep learning (DL) models are widely used in AIoT applications in smart agriculture?	Identifying common DL models used for the same scope of study and assessing the comparison of few DL models.
RQ3	What are common technologies reported for AIoT application in agriculture? Is there any potential technology to explore for the same application?	Providing the review of current related technology.
RQ4	What is the current trend in terms of the number of studies according to similar work?	Providing the context related to the trend of publication number over few years.

**Table 2 sensors-23-03752-t002:** The details for database search process.

Database	Search Keywords	Number of Articles	URL
WOS	Artificial Internet of Things (AIoT)	157	http://www.isiknowledge.com (accessed on 4 March 2023)
	AIoT in agriculture	18	
SCD	Artificial Internet of Things (AIoT)	133	http://www.sciencedirect.com (accessed on 4 March 2023)
	AIoT in agriculture	64	
IEEE	Artificial Internet of Things (AIoT)	174	https://ieeexplore.ieee.org (accessed on 4 March 2023)
	AIoT in agriculture	14	

**Table 3 sensors-23-03752-t003:** Inclusion and exclusion criteria.

No.	Criteria	Database
*Inclusion*		
1.	Articles published between 2017 and 2022.	WOS, SCD, IEEE
2.	Peer-reviewed primary articles.	WOS, SCD, IEEE
3.	Studies within the context of AIoT applied in smart agriculture based on research scope established.	WOS, SCD, IEEE
4.	Articles published in English.	WOS
*Exclusion*		
1.	Secondary or tertiary studies within the context of AIoT applied in smart agriculture based on research scope established.	-
2.	Other document types, such as short articles, proceedings, books etc.	WOS, SCD, IEEE
3.	Redundant studies by the same researcher.	WOS, SCD
4.	Articles published prior to 2017.	WOS, SCD, IEEE

**Table 4 sensors-23-03752-t004:** Quality assessment on the articles based on the questionnaire.

ID	Question
Q1	Are the objectives of the study clearly defined?
Q2	Are the research questions clearly answered?
Q3	Did the study use DL algorithm in its research scope?
Q4	Did the study report a well-described experiment?
Q5	Does the finding of the study prove the validity which relevant to it?

**Table 5 sensors-23-03752-t005:** Quality evaluation.

ID	Research Scope	Author	Final Score
A1	Smart aquaculture farm management system	[33]	3.5
A2	Plant growth monitoring and environmental	[34]	4.0
A3	Algae culture monitor	[35]	5.0
A4	Agriculture 4.0	[36]	4.5
A5	Pest Detection	[31]	3.5
A6	Smart Livestock Surveillance	[37]	3.5
A7	Empowering Things with Intelligence: A Survey	[29]	3.0
A8	AI-Rooted IoT System Design Automation	[38]	3.0

**Table 6 sensors-23-03752-t006:** Number of articles reported for AI and IoT in smart agriculture.

Database	Search Keywords	Number of Related Articles	URL
WOS	Artificial Intelligence in agriculture	1237	http://www.isiknowledge.com (accessed on 4 March 2023)
	IoT in agriculture	319	
SCD	Artificial Intelligence in agriculture	21,309	http://www.sciencedirect.com (accessed on 4 March 2023)
	IoT in agriculture	7045	
IEEE	Artificial Intelligence in agriculture	4690	https://ieeexplore.ieee.org (accessed on 4 March 2023)
	IoT in agriculture	2630	

**Table 7 sensors-23-03752-t007:** Deep learning for smart agriculture.

Method	Description	Data	Disadvantages	Advantages	References
Gated recurrent unit (GRU) and reversible automatic selection normalization (RASN)	A hybrid deep learning method embedded in IoT system as a predictor.	Humidity, wind, and temperature.	Short time data period for prediction	More accurate prediction on humidity, temperature, and wind speed compared to other methods	[39]
Deep learning algorithm on fog nodes (DLEFN)	Deep learning algorithm that performs some DL tasks on fog nodes	Smart agriculture application	No report on prediction and algorithm performance	Efficient resource utilization and able to reduce network congestion	[40]
K-nearest neighbors algorithm (KNN), artificial neural network (ANN), long short-term memory (LSTM), recurrent neural network (RNN), and ensemble subspace discriminate analysis (ESDA)	Proposed method for system monitoring of fruits, smart agriculture of UAVs, crop disease prediction, leaf identification and classification in real-time.	Growth of strawberries, weeds, crops, lettuce production, plant disease, bird species, apple leaf diseases, and agricultural machinery	There is no standard procedure to conduct verification and validation of the proposed method.	Most of the proposed method and algorithms report competitive performance in terms of classification accuracy, identification, and prediction, which applied in computer vision.	[41,42,43,44,45,46,47,48,49,50]
Deep reinforcement learning	Deep reinforcement learning in smart systems	Water consumption	No validation and experiment report	Efficient integrated system incorporating cloud computing to improve food production	[51]

**Table 8 sensors-23-03752-t008:** Previous reports on traditional machine learning methods in smart agriculture.

Method	Description	Data	Disadvantages	Advantages	References
Fuzzy logic controller	Small-scale machine intelligence for smart agriculture	Cultivated data	Only focus on irrigation system	Real time classification with 90% and higher accuracy	[52]
Neural network (ANN) and neural network (CNN)	Application of computer vision and crop farming algorithms for smart farming	Fruit grading and spraying system	No report on the performance based on ANN, CNN, and SVM	Provide direction and trend on using various methods in smart agriculture.	[53,54]
Supervised learning algorithms	An algorithm for detecting and preventing spread of diseases to the whole crop	High-yield crop	Limited amount of data for the trained model and challenging to be used in real-world practice	Classification performance achieved 98% with adequate computation time.	[55]
Support vector, machine, random forest, neural network, and K-nearest neighbors (KNN)	IDS system in smart agriculture and Intelligent biochemical spraying	Field’s water supply and crop/weed	Uses standard machine learning algorithms, and the performance is not tested with deep learning.	Focusing security IDS on smart farming and compare with several algorithms (SVM achieve higher accuracy).	[56,57,58]
Wrapper PART (WPART)	Machine learning algorithm for smart farming	Crop productivity	No time series analysis for prediction future value from previously observed values	WPART able to achieve high accuracy with 90% and above in crop productivity and drought prediction	[59]
Various machine learning algorithms	Application and suggestion in smart agriculture using various machine learning algorithms	Cultivate crops productivity	Challenging to non-technical people, specifically when introduced to machine learning algorithms	Direct recommendation to individual and farmer in managing their crop cultivation	[60]

## Data Availability

Not applicable.

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
