# Peer review of "Recent Advancements and Challenges of AIoT Application in Smart Agriculture: A Review"

_sensors, 2023, doi:10.3390/s23073752_

Round 1

Reviewer 1 Report

Some suggestions for further improvement of the manuscript:

1. This article cites several articles and gives a detailed description, but it is recommended to add the difference between different methods and compare the advantages and disadvantages.

2. Suggest adding your own opinion to other people's methods.

3. Lack of conclusions.

Author Response

Dear Reviewer,

Thank you for your constructive comments previously. We have made major revision on our article, please check our cover letter. 

Thank you. 

Reviewer 2 Report

Summary:  In this paper, several promising AIoT applications are discussed to further improve the concepts, applications, and user benefits associated with AIoT ( artificial intelligence of things). The challenges to the adoption of AIoT technology in modern agriculture are also discussed.

Suggestions and Comments:

- Pay attention to English Mistakes: e.g., "user benefits *associate* with AIoT"

- The authors need to provide more details about their contribution in the abstract.

- In addition, the authors are invited to summarize their contributions in the form of a list of short sentences in the introduction.

- At the end of the introduction, it is necessary to add a short paragraph that describes the structure of the paper.

- In addition, the authors may add a Figure which illustrates graphically the structure of the paper.

- A new section about the search methodology adopted in this work is needed (search queries, databases, inclusion/exclusion criteria, etc.)

- A comparison with similar survey papers will be useful too.

- The following interesting works (and others) need to be included in the list of references covered by this study:

1. https://www.sciencedirect.com/science/article/pii/S2772375522000156

2. sciencedirect.com/science/article/pii/S221478532101052X

3. https://www.sciencedirect.com/science/article/pii/S2772375522000181

- The authors also need to report on the use of smartphone sensors for collecting/analyzing data and taking decisions.

- For this purpose, they may consider the following interesting references (and others):

1. https://www.hindawi.com/journals/js/2015/195308/

2. https://link.springer.com/chapter/10.1007/978-981-19-1906-0_56

- The authors need to report on the use of formal techniques for checking the correctness of AI-based solutions. For this purpose, the following references may be included:

1. https://ieeexplore.ieee.org/document/9842406

2. https://arxiv.org/abs/2104.02466

3. https://link.springer.com/chapter/10.1007/978-3-030-25540-4_25

- The content of some tables needs to be summarized and presented in a more concise manner.

- The paper may be enriched with more figures and tables which summarize the presented findings.

- A discussion and conclusion sections are missing.

- The authors need to provide recommendations for the use of specific types of AI for each specific need in modern agriculture.

- Economic and environmental aspects need to be considered too.    

Author Response

Dear Reviewer,

Thank you for your constructive comments previously. We have made major revision on our article, please check our cover letter. 

Thank you. 

Dear Reviewer,

Thank you for your constructive comments previously. We have made major revision on our article, please check our cover letter. 

Thank you. 

Round 2

Reviewer 2 Report

The authors covered all my comments and suggestions. Good luck.